# Follistatin-like 1 and Biomarkers of Neutrophil Activation Are Associated with Poor Short-Term Outcome after Lung Transplantation on VA-ECMO

**DOI:** 10.3390/biology11101475

**Published:** 2022-10-08

**Authors:** Cecilia Veraar, Enzo Kirschner, Stefan Schwarz, Peter Jaksch, Konrad Hoetzenecker, Edda Tschernko, Martin Dworschak, Hendrik J. Ankersmit, Bernhard Moser

**Affiliations:** 1Department of Anesthesiology, Intensive Care Medicine and Pain Medicine, Division of Cardiothoracic and Vascular Anesthesia and Intensive Care Medicine, Medical University of Vienna, 1090 Vienna, Austria; 2Applied Immunology Laboratory, Medical University of Vienna, 1090 Vienna, Austria; 3Department of Thoracic Surgery, Medical University of Vienna, 1090 Vienna, Austria

**Keywords:** primary graft dysfunction, lung transplantation, extracorporeal membrane oxygenation, neutrophil activation, follistatin-like 1, lipocalin 2, chemokine 4, sequential organ failure assessment score

## Abstract

**Simple Summary:**

The determination of biomarkers linked to adverse outcome after lung transplantation is crucial to identify vulnerable recipients and to define targeted therapeutic options. In this study, we investigated the perioperative course of a cytokine that has been related to graft-versus-host disease named Follistatin-like 1 and cytokines associated with neutrophil activation. We included 42 consecutive patients with different etiologies of end-stage pulmonary disease undergoing double lung transplantation. We found that all cytokines increased immediately after surgery. Follistatin-like 1 was significantly reduced at baseline in patients developing primary graft dysfunction. Serum concentrations of neutrophil-derived cytokines were related to prolonged extracorporeal membrane oxygenation and increases sequential organ failure assessment. Follistatin-like 1 serum concentrations seem exhausted in patients developing primary graft dysfunction. This finding could be of great importance, especially since this protein appears to contribute to allograft tolerance. However, larger trials are necessary to assess the predictive power of each of these cytokines.

**Abstract:**

The investigation of biomarkers associated with undesired outcome following lung transplantation (LuTX) is essential for a better understanding of the underlying pathophysiology, an earlier identification of susceptible recipients and the development of targeted therapeutic options. We therefore determined the longitudinal perioperative course of putative cytokines related to neutrophil activation (chemokine CC motif ligand 4 (CCL-4), interleukin (IL)-23 and Lipocalin 2 (LCN2)) and a cytokine that has been implicated in graft-versus-host disease (Follistatin-like 1 (FSTL1)) in 42 consecutive patients undergoing LuTX. We plotted receiver-operating curves (ROC) to assess the predictive power of the measured cytokines for short-term outcomes namely primary graft dysfunction (PGD), early complications requiring extracorporeal membrane oxygenation (ECMO), and a high postoperative sequential organ failure assessment (SOFA). All cytokines increased immediately after surgery. ROC analyses determined significant associations between CCL4 and a high SOFA score (area under the curve (AUC) 0.74 (95%CI:0.5–0.9; *p* < 0.05), between LCN2 and postoperative ECMO support (AUC 0.73 (95%CI:0.5–0.9; *p* < 0.05), and between FSTL1 and PGD (AUC 0.70 (95%CI:0.5–0.9; *p* < 0.05). The serum concentrations of the neutrophil-derived cytokines LCN2 and CCL4 as well as FSTL1 were all related to poor outcome after LuTX. The specific predictive power, however, still has to be assessed in larger trials. The potential role of FSTL1 as a biomarker in the development of PGD could be of great interest particularly since this protein appears to play a crucial role in allograft tolerance.

## 1. Introduction

Primary graft dysfunction (PGD) is a syndrome that encompasses a spectrum of mild to severe lung injury that occurs within the first 72 h after lung transplantation (LuTX) [1]. PGD is the leading cause of early morbidity and mortality in LuTX recipients and the main reason for developing chronic lung allograft dysfunction (CLAD), the major factor for reduced long-term survival [2,3]. Early performance and lung function are usually assessed by a standardized scoring system of the International Society for Heart and Lung Transplantation (ISHLT). The PGD grading system was developed according to the acute respiratory distress syndrome (ARDS) classification and was proposed for the first time in 2005 and revised in 2017 [4,5]. PGD is characterized histologically by diffuse alveolar damage, but is graded clinically combining the PaO_2_/FiO_2_ (P/F) ratio with the presence of radiographic infiltrates [1].

The current understanding of the pathogenesis of PGD emphasizes multiple pathways involving inflammation, innate immunity-particularly neutrophil activation, as well as platelet and coagulatory dysfunction. These pathways subsequently induce endothelial and alveolar epithelial injury in the newly transplanted lungs [6]. Especially interleukin (IL)-6, IL-8 and receptor for advanced glycation endproducts (RAGE) have shown promising results in multiple heterogeneous samples for identifying patients at risk for PGD. IL-8 is a chemokine that is released by multiple cell types during lung reperfusion to recruit and activate neutrophils at the site of injury within the lung [7]. Another study found that neutrophil extracellular traps (NETs), which were released by neutrophils upon activation, were increased in bronchioalveolar fluid, but not in blood samples within 24 h after transplantation in recipients who developed PGD [8]. 

Lipocalin 2 (LCN2) is constitutively expressed and stored in neutrophilic granules which makes it a possible biomarker for neutrophil activity. We previously reported significantly increased LCN2 serum concentrations in patients with early onset CLAD and in bronchial epithelium of BOS and RAS patients [9]. Chemokine CC motif ligand 4 (CCL4) is released from hypoxic alveolar macrophages and promotes neutrophil trafficking into the injured lung [10]. Cantu et al. found enhanced expression levels of CCL4 in PGD patients [11]. In addition, our study group observed decreased release of CCL4 and LCN2 in serum of patients undergoing open-heart surgery with continued mechanical ventilation during cardiopulmonary bypass (CPB) [12,13,14].

Furthermore, the IL-23/IL-17 axis has been implicated to play a pivotal role in neutrophil migration in patients with rheumatoid arthritis [15]. IL-23 is known to be rapidly produced by activated macrophages or dendritic cells at the site of infection. IL-23 subsequently activates Th17/Th IL-17 cells and other IL-17–producing cells whereby IL-17 induces G-CSF production from stromal cells. The IL-23/IL-17/G-CSF pathway subsequently augments neutrophil recruitment to the infection site [16]. Levels of IL-23 and IL-17 cytokine serum concentrations have been linked to PGD disease severity [17].

An animal model in rats with bronchiolitis obliterans (OB) after orthotopic tracheal transplantation showed that anti-IL-23 blockade inhibits IL-17 production and thereby protecting allograft rejection and the development of CLAD [18].

Follistatin-like 1 (FSTL1) also known as follistatin-related protein is a secreted protein involved in the respiratory development and regulation of immunological processes [19,20]. FSTL1 has been introduced as a new inflammatory mediator which regulates several actions of inflammatory cells. In addition, FSTL1 is known to be a sensitive marker to identify oxidative stress. Clinical research found that FSTL1 rose in serum of chronic obstructive pulmonary disease (COPD) patients combined with pulmonary hypertension and played a pivotal role during airway remodelling in the asthmatic patient. [19,21,22]

Currently, there is a lack of knowledge on the impact of neutrophil activation and oxidative stress on short-term outcomes such as PGD and early postoperative complications (i.e., rates for re-intubation, hemofiltration, re-exploration, and organic brain syndrome (OBS)), high postoperative sequential organ failure assessment (SOFA) and 30-day mortality. 

We therefore investigated the longitudinal perioperative course of cytokines putatively implicated in PGD (CCL-4, IL-23, LCN2, and FSTL1) and analyzed their impact on PGD, early complications, high SOFA scores (>13 points), and 30-day mortality. In addition, we performed long-term follow up focusing on CLAD and 3-year mortality. Since PGD is associated with blood product administration we also analyzed FSTL1 concentrations in packed red blood cells (PRBC) and fresh frozen plasma (FFP).

## 2. Materials and Methods

The study was conducted in accordance with the Declaration of Helsinki (as revised in 2013) and was approved by the Ethics Committee of the Medical University of Vienna (EK1363/2018). Written informed consent was obtained from all study participants prior to inclusion.

### 2.1. Study Design, Setting, and Patients

The study was designed as an explorative prospective cohort study combining clinical and experimental research. It was performed at the Medical University of Vienna. 

We included 42 consecutive patients with end-stage pulmonary disease [COPD (n = 15), cystic fibrosis (CF) (n = 12), idiopathic pulmonary fibrosis (IPF) (n = 6), idiopathic pulmonary arterial hypertension (IPAH) (n = 4), CLAD (n = 6)] undergoing LuTX on veno-arterial extracorporeal membrane oxygenation (VA-ECMO) from May 2018 until April 2019. We excluded patients who were younger than 18 years and patients who did not consent. Blood samples were drawn before surgery, at ICU admission, and once on each of the three following postoperative days (PODs) as well as on POD5. 

### 2.2. Sandwich ELISA Technique 

LCN2, IL-23, CCL-4, and FSTL1 serum concentrations and FSTL1 concentrations in 7 units of PRBC and 4 units of FFP were measured by commercially available enzyme-linked immunosorbent assay (ELISA) kits (R&D Systems, Minneapolis, MN, USA) according to the manufacturers’ instructions. 96-well microplates were incubated with capture antibodies (mouse anti-human LCN2, IL-23, CCL-4, and FSTL1) overnight at room temperature. Blocking was carried out with assay buffer. After incubation with samples and washing, HRP-conjugated detection antibodies were added biotinylated goat anti-human (LCN2, IL-23, CCL-4, and FSTL1). A color reaction was obtained with peroxidase reagent tetramethylbenzidine (TMB) (Sigma-Aldrich Corp., St. Louis, MO, USA) and the optical density (OD) was read at 450 nm using an absorbance microplate reader for ELISA, the Infinite F50 (Tecan, Männedorf, Switzerland). We calculated the inter-assay and intra-assay coefficient of variability (%CV). 

### 2.3. Perioperative Management 

#### 2.3.1. Anesthesia and Surgical Procedure

All patients received standard perioperative monitoring after entering the operating room. Anesthesia was induced with 2 mg midazolam, fentanyl 2 μg/kg, propofol 2 mg/kg and cis-atracurium 0.2 mg/kg. Antibiotic prophylaxis was administered using piperacillin/tazobactam 4.5 g 30–60 min before skin-incision. A Swan-Ganz catheter and a central venous catheter were placed into the internal jugular vein. Anesthesia was maintained with fentanyl 3 µg/kg/h and propofol 6 mg/kg/h. Transesophageal echocardiography was installed to monitor cardiac function, volume status and vasoactive pharmacological support during surgery. An initial dose of 60 IU/kg of heparin was administered prior to initiation of intraoperative VA-ECMO. Patients were kept normothermic throughout the entire surgical procedure [23]. 

Donor lungs were harvested in the course of a multi-organ procurement, preserved with a colloid-containing low potassium solution and kept inflated during transport. Each donor lung was classified according to the Oto score, known as a grading system, which encompasses five variables that are routinely available including age, smoking history, chest X-ray, secretions and PaO_2_/FiO_2_ ratio. Each variable receives 0 to 3 points, based on clinical importance. The PaO_2_/FiO_2_ ratio was weighted from 0 to 6. The Oto score ranges in total from 0 (ideal lungs) to 18 (the worst possible) [24].

Bridging strategies were chosen according to patients’ preoperative hemodynamic and respiratory conditions. Patients with severe hypoxic respiratory failure received a VV-ECMO (Cardiohelp, Oygenator Quadrox)/(Xenios, Oxygenator Hilite) with a 2-site (femoro-jugular) or a single-site large-bore double-lumen cannula ranging from 27F to 31F (Avalon Laboratories, Los Angeles, CA, USA) [25]. LuTX was performed through bilateral thoracotomy or via clamshell incision and intraoperative central VA-ECMO support with or without extension into the postoperative period. In patients requiring prolonged VA-ECMO, central cannulation was switched to a femoro-femoral access after implantation of the lungs. All patients were transplanted on intraoperative VA-ECMO (Medtronic Carmeda) with a heparin-coated tubing, hollow-fiber oxygenator (Medtronic Inc., Minneapolis, MN, USA), centrifugal pump (Biomedicus), flow probes and a 3/8-inch internal diameter. 

For prolonged femoro-femoral VA-ECMO, a 17-19F drainage and 15-17F reperfusion cannulas were inserted (Bio-Medicus Cannula, Medtronic Inc., Minneapolis, MN, USA). Subcutaneous low molecular-weight heparin was administered at a therapeutic dosage for anticoagulation during postoperative ECMO support. VA-ECMO support was terminated when patients showed a stable hemodynamic profile, had normal chest X-rays, adequate oxygenation (FiO_2_ < 0.5), requiring only low or moderate respiratory support by the ventilator and an even fluid balance [26]. 

#### 2.3.2. PGD Grading

PGD grading was performed in accordance with the latest guidelines of the ISHLT working group [4,13]. Patients were graded on the basis of the Horowitz index (PaO_2_/FiO_2_ ratio) and chest radiographs at time points T0, T24, T48, and T72 h. Time point T0 was measured 2 h after arrival at the ICU after stabilization of patient’s respiratory and hemodynamic conditions, time points T24, T48 and T72 were performed 24 h, 48 h and 72 h after arrival at the ICU, respectively. Experienced thoracic radiologists evaluated the chest radiographs. Extubated patients were not assigned to PGD grades but were listed separately [27].

#### 2.3.3. Postoperative Immunosuppressive Regimen 

In this case, 84% of all investigated patients received induction therapy with alemtuzumab, anti-thymocyte globulin was employed in 3% while 13% did not receive induction therapy directly after ICU admission. Maintenance therapy consisted of tacrolimus, mycophenolat-mofetil (only in patients who did not receive alemtuzumab) and corticosteroids. All patients received postoperative anti-infectious prophylaxis with piperazillin/tazobactam. Lifelong pneumocystis prophylaxis was initiated with intravenous trimethoprim- sulfamethoxazole and prophylactic inhalation therapy with amphotericin B and gentamicin. CMV prophylaxis was performed with CMV hyperimmuno-globulines (on POD 1, 7, 14, and 21) and valganciclovir. 

#### 2.3.4. Definition of SOFA

SOFA was assessed every 24 h in all patients after ICU admission. The SOFA score was calculated by assessing the degree of dysfunction of the following five organ systems, i.e., respiratory, coagulation, hepatic, cardiovascular and renal, from 0–4 points. The assessment of the neurological function using the Glasgow coma scale (GCS) was excluded due to limited evaluability in actively sedated patients. The following data was collected: need for mechanical ventilation, PaO_2_/FiO_2_ ratio, platelet count, bilirubin, mean arterial pressure, doses of adrenergic agents, creatinine, and urine output [23,28].

### 2.4. Statistical Analysis 

Data are presented using descriptive statistics. Mean ± standard deviation (SD) or median (25th percentile, 75th percentile) was calculated for continuous variables depending on their specific distribution. Categorical variables are given as frequency (percentage). The chi-square test was used for categorical variables. The student’s paired *t*-test and the Wilcoxon rank-sum test was applied for continuous variables with parametric and non-parametric paired groups. The Mann-Whitney U test was performed for non-parametric unpaired t-testing. The Friedman test was applied for multiple comparisons of repeated measures. In addition to reporting absolute serum cytokine concentrations as measured we also determined relative changes from the respective baseline levels depicted as x-relative increases (x RI). We further plotted the receiver operating characteristic (ROC) curve and calculated the area under the curve (AUC) for all cytokines to assess the predictive power for PGD, postoperative ECMO support and a high SOFA score (>13 points). We calculated the Youden Index only in those cytokines in which the AUC was statistically significant for a specific condition. We assessed optimal cut-off values for FSTL1, LCN2 and CCL4 RI between baseline levels and concentrations measured at the time of ICU admission to perform binary logistic regression and evaluate the sensitivity, specificity, positive predictive value (PPV) and negative predictive value (NPV). The odds ratio (OR) was computed using binary logistic regression analysis. The level of statistical significance was set at 0.05 (for two-tailed tests). Statistical analyses and visualization were performed using SPSS software (v27; IBM SPSS Inc., Chicago, IL, USA) and GraphPad Prism 9 (GraphPad Software, La Jolla, CA, USA). Boxplots were designed as followed: Box: signifies 1st to 3rd quartiles, Bar: medians, Whiskers: min to max, showing all individual values as dots. 

## 3. Results

### 3.1. Demographic Data

Demographic data and perioperative characteristics are depicted in Table 1. Five patients underwent Re-LuTX due to CLAD. Of these CLAD patients three were initially diagnosed with CF, one with IPF and one patient had IPAH. Four patients undergoing LuTX, diagnosed with COPD, CF, IPF, and IPAH were bridged to transplantation via VV-ECMO and all received intraoperative VA-ECMO of whom one received prolonged VA-ECMO after surgery. Modified PGD grading according to the ISHLT consensus definition of 2017 of the patients at 0 h, 24 h, 48 h, and 72 h after surgery is shown in Table 2 [13]. Patients requiring prolonged postoperative ECMO support and presenting pulmonary infiltrates on chest X-ray were graded as PGD 3. Patients on prolonged ECMO support without pulmonary infiltrates were deemed ungradable [13].

In this case, 14 patients met the definition for PGD after surgery and 15 patients’ 24 h after surgery. Seven patients developed PGD postoperatively without the need of ECMO support. 

#### 3.1.1. Perioperative Course of CCL4, IL-23, LCN2, Leucocytes, and FSTL1 Serum Concentrations

CCL4 serum concentrations increased significantly during surgery [median 26.6 pg/mL (16.9, 31.7) to 85.9 pg/mL (51.6, 352.9); *p* < 0.05] and decreased steadily thereafter (Figure 1A). During the same period IL-23 serum concentrations climbed as well [median 6.1 pg/mL (0.0–152.8) to 222.1 pg/mL (63.1–509.5); *p* < 0.05] and decreased slowly within the following days (Figure 1B). LCN2 increased significantly from baseline to end of surgery [median 165.6 pg/mL (92.7, 262.8) to 262.8 (149.0, 375.7); *p* < 0.05] and decreased steadily thereafter as shown in Figure 1C. In patients diagnosed with PGD, FSTL1 serum concentrations increased statistically significant from baseline to the end of surgery [median 8.0 pg/mL (2.4, 61.8) to 79.5 (59.4, 89.6); *p* < 0.05] and decreased significantly thereafter, from end of surgery to POD 1 [median 79.5 (59.4, 89.6) to 25.9 (16.1, 46.2); *p* < 0.05] (Figure 1D). FSTL1 serum concentrations at baseline were significantly reduced in future PGD patients compared patients without PGD [median 2.4 pg/mL (0.79, 6.2) and 15.6 pg/mL (4.5, 87.1); *p* < 0.05] but not compared to healthy volunteers [median 6.3 pg/mL (3.2, 39.4); *p* > 0.05] (Figure 1E). In contrast, at end of surgery there was a significant difference between PGD and no PGD compared to healthy volunteers, but not between PGD and noPGD patients [median 81.7 pg/mL (67.8, 90.2) and 78.4 pg/mL (58.9, 85.9) compared to 6.3 pg/mL (3.2, 39.4); *p* < 0.05] (Figure 1F). 

Due to the high proportion of fibrotic patients developing PGD we compared FSTL1 serum concentrations between patients with fibrotic and non-fibrotic pulmonary disease. At baseline, after surgery and RI of FSTL1 was not significantly different in non-fibrotic compared to fibrotic patients [median 8.9 ng/mL (2.0–71.3) and median 6.2 ng/mL (2.7–71.6); *p* = 0.97], [median 76.7 ng/mL (62.4–83.4) and median 83.4 ng/mL (41.5–96.7); *p* = 0.45] and [median 7.5 RI (0.4–29.3) *p* = 0.83 and median 4.3 RI (0.0–26.8)].

#### 3.1.2. Greater Relative Increase of FSTL1 in Patients with PGD

Unpaired nonparametric t-testing revealed a statistically significant increase in RI of FSTL1 serum concentrations in patients with PGD compared to patients without PGD [median 13.5 FI (3.4–35.5) and 3.9 RI (0.0–21.2); *p* < 0.05] (Figure 2A). In addition, after exclusion of patients requiring ECMO, FSTL1 RI was still significantly higher in PGD patients [median 29.7 RI (5.9–85.4) and median 4.3 RI (0.0–27.0); *p* < 0.05] (Figure 2B). We did not observe a significant difference in FSTL1 RI between patients with and without postoperative ECMO support [median 7.4 RI (0.1–30.9) and 7.0 RI (0.5–12.8); *p* < 0.05] (Figure 2C). LCN2 RI was significantly increased in patients with prolonged ECMO compared to patients without ECMO support [median 1.7 RI (0.4–3.2) and 0.3 RI (−0.0–1.1); *p* < 0.05] (Figure 2D). The same applied for patients with postoperative complications comprising re-intubation, hemofiltration (HF), OBS, re-thoracotomy or died within 30 days compared to patients without such complications [median 0.9 RI (0.3–2.2) and 0.2 RI (0.3–2.2); *p* < 0.05] (Figure 2E). There was no statistically significant difference in RI of IL-23, CCL4 and LCN2 between PGD and patients without PGD as depicted in the Appendix A. 

#### 3.1.3. The Associations between FSTL1, CCL4, LCN2 and Postoperative Outcome

Three conditions were formulated for the ROC analyses. Youden Indices were calculated to define optimal cut-off values for FSTL1, CCL4 and LCN2 FI to predict PGD, postoperative ECMO support and a high SOFA score. For the condition PGD, the AUC of FSTL1 was statistically significant with 0.7 (95% CI: 0.5–0.9; *p* < 0.05) as shown in Figure 3A. We obtained a sensitivity of 65%, specificity of 81%, PPV of 74% and NPV of 75% to predict PGD; calculated with a cut-off value of 10.2 FSLT1 FI. For CCL4, LCN2 and IL-23 the AUC was 0.6 (95% CI: 0.3–0.7; *p* > 0.05), 0.6 (95% CI: 0.3–0.7; *p* > 0.05) and 0.6 (95% CI: 0.3–0.8; *p* > 0.05), respectively. 

For the condition PGD the AUC of FSTL1 was 0.8 (95% CI: 0.5–0.9; *p* < 0.05) after exclusion of ECMO patients as shown in Figure 3B. 

With postoperative ECMO as the condition, we found a statistically significant association for LCN2 with an AUC of 0.7 (95% CI:0.5–0.9; *p* = 0.04), but not for IL-23, CCL4, and FSTL1 [AUC 0.4 (95% CI: 0.0–0.7; *p* > 0.05), 0.46 (95% CI: 0.2–0.6; *p* > 0.05), and 0.55 (95% CI: 0.3–0.7; *p* > 0.05)], respectively; as visualized in Figure 3C. To predict the need of postoperative ECMO support using a cut-off value of 0.3 LCN2 FI, we achieved a sensitivity of 87%, specificity of 50%, PPV of 30%, and NPV of 94%. 

For the condition high SOFA score there was a statistically significant association for CCL4 with an AUC of 0.7 (95% CI: 0.5–0.9; *p* < 0.05) and for FSTL1 with an AUC of 0.8 (95% CI: 0.6–0.9; *p* < 0.05) as depicted in Figure 3D. To predict a high SOFA score at ICU admission a cut-off value of 9.2 CCL4 FI was applied, thereby reaching sensitivity of 66%, specificity of 77%, PPV of 33% and NPV of 93%. However, this was not the case for LCN2 and IL-23 [AUC 0.5 (95% CI: 0.1–0.8; *p* > 0.05), 0.2 (95% CI: 0.0–0.3; *p* > 0.05)].

#### 3.1.4. Donor, Recipient and Procedure Related Risk Factors Associated with PGD

Donor related factors associated with PGD comprising donor age, type of donor such as donation after cardiac death (DCD) or donation after brain death (DBD), smoking status and high Oto score (>13 points) did not reveal statistically significance criteria to identify patients at risk for PGD. Regarding recipient related factors, COPD patients were significantly less likely to develop PGD compared to patients with other end stage pulmonary disorders [OR 0.16 (95% CI: 0.0–0.8; *p* < 0.05)]. Procedure related factors associated with PGD such as transfusion of more than six units of packed red blood cells (PRBC)s, random donor platelets (RDP) and high FSTL1 RI (>10.2) occurred more frequently in PGD patients compared to patients without PGD [OR 7.1 (95% CI:1.5–32.0); *p* < 0.05)], OR 4.2 (95% CI:1.1–16.4); *p* < 0.05] and OR 8.2 (95% CI:1.8–35.9); *p* < 0.05)] (Figure 4).

#### 3.1.5. FSTL1 Concentrations Measured in PRBCs and FFPs

We measured FSTL1 concentrations of 7 expired and non-expired PRBCs and 4 expired FFPs. We found only traces of FSTL1 in 3 of 7 investigated PRBCs. In this case, 11 ng/mL were measured in one non-expired and 3 and 6 ng/mL were found in two expired PRBCs, respectively. We further detected FSTL1 concentrations of 1, 7, 29, and 30 ng/mL in the investigated FFPs, respectively.

#### 3.1.6. Short-Term and Long-Term Outcome of Patients Undergoing LuTX

Short- and long-term outcome is shown in Table 3. Overall, six patients had a high SOFA score >13 after LuTX. In this case, 14 patients (amongst them seven CF patients) developed PGD. Only one patient was diagnosed with antibody-mediated rejection, and none had acute cellular rejection. Two COPD patients had to be re-intubated within the observational period of five days. Moreover, six patients developed organic brain syndrome (OBS) and required corresponding medical treatment. Five patients needed renal replacement therapy. Furthermore, two patients died within 30 days after LuTX [23].

Regarding long-term outcome, seven patients developed CLAD, three of them were diagnosed with BOS, and four patients presented a mixed CLAD phenotype. In total, 14 patients died within three years post transplantation. Six patients died due to infection, two patients due to CLAD, one patient due to bowel ischemia, two patients after re-transplantation and in three patients the cause was unknown. 

## 4. Discussion

PGD has significant impact on early morbidity and mortality of patients undergoing LuTX. Understanding the underlying pathophysiology is therefore crucial to develop novel strategies to identify and reduce PGD risk.

In this study we observed significantly reduced FSTL1 serum concentrations in future PGD patients prior to LUTX compared to patients who did not develop PGD. In response to transplantation, FSTL1 increased more extensively in these patients as compared to patients who did not develop PGD. Immediately after surgery, FSTL1 serum concentrations dropped back to baseline levels in both groups. These findings suggest that preoperative factors may limit an adequate FSTL1 production and therefore put these patients at an increased risk for PGD. In our cohort, PGD patients were mainly diagnosed with fibrotic disorders comprising IPF and CF. Research reported that in the early stage of fibrotic diseases, FSTL1 serum concentrations were up regulated [29]. In contrast, at baseline FSTL1 production seemed already exhausted in patients with end-stage pulmonary fibrosis developing PGD but could still be temporarily increased by the intraoperative inflammatory response.

Additionally, we identified an intense intraoperative neutrophil activation, resulting in elevated serum concentrations of CCL4, LCN2, and IL-23 immediately after surgery. CCL4 was linked to a severe inflammatory reaction, resulting in high SOFA scores. Raised LCN2 levels were found in patients requiring postoperative ECMO support and in patients with early complications comprising re-intubation, hemofiltration, revision, and OBS. 

In the literature several modifiable and uncontrollable risk factors for PGD have been identified which can be further divided into donor, recipient and procedure related factors. Several studies analyzed the impact of large-volume PRBC transfusion on PGD. In a prospective multicenter cohort study on 1225 patients, transfusion of more than 1 L of PRBCs was associated with a nearly 2-fold increased risk of developing PGD grade 3 [30]. Menger and colleagues even found in the multivariate analysis that transfusing more than 5 units of PRBCs was significantly associated with mortality in LuTX patients [31]. In order to elucidate whether FSLT1 is introduced via transfused blood products we measured FSTL1 concentrations in PRBCs and FFPs. We found only traces of FSTL1 in 3 of 7 investigated PRBCs and FSTL1 levels, comparable to serum concentrations of healthy volunteers in all investigated FFPs. Since, PRBC transfusion was not associated with high FSTL1 RI, we assume that increased RI in PGD patients was not induced by blood product transfusion. Donor-specific blood transfusion before transplantation, however, can obviously beneficially mediate allograft tolerance via a FSTL1 pathway [32]. 

In addition, another study reported that extracorporeal circulation does not appear to modify the serum level of circulating FSTL1 protein as well [33]. 

PGD has consistently and independently been associated with BOS, a crucial limiting factor for long-term survival after LuTX [34]. Recently, it has been shown that RAGE, a marker of increased epithelial injury in the course of PGD, correlates with BOS, indicating a link between the degree of epithelial injury and aberrant repair or on-going immune activation leading to BOS [35]. In contrast to RAGE, FSTL1 in our cohort was neither significantly elevated in BOS patients nor in PGD patients who developed CLAD during follow-up whereby it has to be taken into account that only two of our PGD patients developed CLAD during the observational period of three years. 

Tissue damage of the lungs when subjected to ischemia is further aggravated by re-oxygenation during reperfusion [36]. Ischemia/reperfusion injury is discussed as one of the major causes for PGD and is directly related to the formation of reactive oxygen species, endothelial cell injury, increased vascular permeability, and activation of neutrophils and platelets, cytokines, and the complement system [37]. The concentration of derivatives of reactive oxygen metabolites (dROMs) can be a useful marker for reactive oxygen species [38]. Recently, a study in healthy men reported that plasma FSTL1 concentrations independently correlated with dROMs and the authors concluded that FSTL1 serves as an indicator for oxidative stress-related disorders [39].

The role of FSTL1 remains controversial. Pro- and anti-inflammatory actions were described in the literature. Murakami et al. proposed that two signaling pathways of FSTL1 might explain the pro- and anti-inflammatory effects of FSTL1. The study group suggested that FSTL1 may exert its pro-inflammatory effect through CD14 and Toll-like receptor 4 pathways, whereas the anti-inflammatory effect involves the disco interacting protein 2 homolog A (DIP2A) pathway [40]. Several studies reported on anti-inflammatory mechanisms of FSTL1 during cardiac surgery in an animal model. Furthermore, FSTL1 was overexpressed in rats tolerating their newly implanted allografts in contrast to those who showed rejection. Moreover, induced overexpression of FSTL1 via adenovirus gene transfer in vivo significantly prolonged allograft survival in association with inhibition of pro-inflammatory cytokines [32]. Another investigation found that administration of human FSTL1 protein significantly attenuated myocardial infarct size in a mouse and a pig model of ischemia/reperfusion, which was associated with diminished apoptosis and inflammatory responses in the ischemic heart [41]. These findings are in line with our study. Increased FSTL1 serum concentrations prior to surgery seem protective against PGD. The insult of ischemia and reperfusion during LuTX can induce a short enhanced FSTL1 excretion, reaching levels of healthy volunteers. However, the duration of FSTL1 secretion seems limited in these patients without preventing the manifestation PGD. In contrast, other trials demonstrated FSTL1′s important role as a pro-inflammatory cytokine in the pathogenesis of bronchial asthma, promoting airway remodeling and airway inflammation in patients with asthma [42]. FSTL1 levels in asthmatic patients’ serum and bronchoalveolar lavage fluid were higher than in healthy controls, and the degree of its augmentation was positively correlated with tracheal airway smooth muscle and reticular basement membrane thickening [42,43]. Proteomic analysis of the sputum of patients with asthma also displayed that FSTL1 is one of the most abundantly expressed proteins [44].

In this study we also observed a strong association between elevated CCL4 serum concentrations and a high SOFA score. As previously reported by our study group, IL-6 was a highly sensitive marker for the detection of patients with high SOFA scores [23]. IL-6 is known as a sensitive mediator of the acute phase reaction and is frequently determined to detect inflammatory conditions in the clinical routine [45]. The role of CCL4 during acute phase reactions has not been investigated previously. However, a recent study on atherosclerosis reported a strong relation between CCL4, IL-6, and tumor necrosis factor α (TNF-α). In apolipoprotein E-deficient mice model, CCL4 antibody treatment reduced circulating IL-6 and TNF-α whereby the authors concluded that CCL4 inhibition seems to be a promising tool within the box of anti-inflammatory therapeutics in patients diagnosed with atherosclerosis [46].

In this study postoperative VA-ECMO has been used to stabilize patients with refractory hypoxemia or those with hemodynamic instabilities. These patients had significantly elevated LCN2 levels. Cardiac surgery on CPB can activate neutrophils, which subsequently release their granular content into the blood stream. These granules are recognized as a major source of LCN2 in plasma [47,48]. Several factors support the association of CPB and LCN2 release. First, CPB is a recognized pro-inflammatory condition which leads to activation of neutrophils [47]. Longer duration of CPB is further associated with higher levels of circulating LCN2 [49]. Moreover, on-pump cardiac surgery has been associated with more profound neutrophil activation compared to off-pump surgery [48,49,50]. LCN2 is also produced by renal cells and accumulates in large amounts in proximal tubular epithelial cells in experimental and clinical ischemic kidney injury [51]. In patients undergoing cardiac surgery, increased concentrations of LCN2 in the blood have been shown to be an early predictor of AKI whereby CPB support is of course a major confounding factor [52]. In our study, LCN2 was significantly increased in patients with postoperative complications including hemofiltration. Median levels of LCN2 were higher in patients requiring hemofiltration, which however did not reach statistical significance most likely due to the low number of affected patients in this cohort.

Our study does have limitations primarily owing to its single-center design and the relatively small sample size that does not permit further subgroup analyses. Moreover, there is an unavoidable bias related to patients’ age and associated comorbidities. Patients diagnosed with COPD and IPF were by far older than CF and IPAH patients. In addition, patients with IPF were more likely to receive ECMO bridging to transplantation. While comparative results (e.g., control vs. PGD) obtained in one test could be repeated in separate ELISA tests, the absolute values for the individual serum samples vary in our experienced hands, revealing an inter-assay %CV of 10.9 and an intra-assay %CV of 6.7 for FSTL1. Therefore, absolute serum concentrations values have to be treated with caution. 

## 5. Conclusions

In this prospective cohort study, we could relate neutrophil-derived cytokines LCN2, CCL4 as well as FSTL1 with poor outcome after LuTX. Exhausted serum concentrations of FSTL1 prior to LuTX may identify future PGD patients. This role of FSTL1 as a biomarker in the development of PGD could be of great interest particularly since this protein appears to play a crucial role in allograft tolerance. However, the predictive power of each of these cytokines still has to be assessed in larger trials.

## Figures and Tables

**Figure 1 biology-11-01475-f001:**
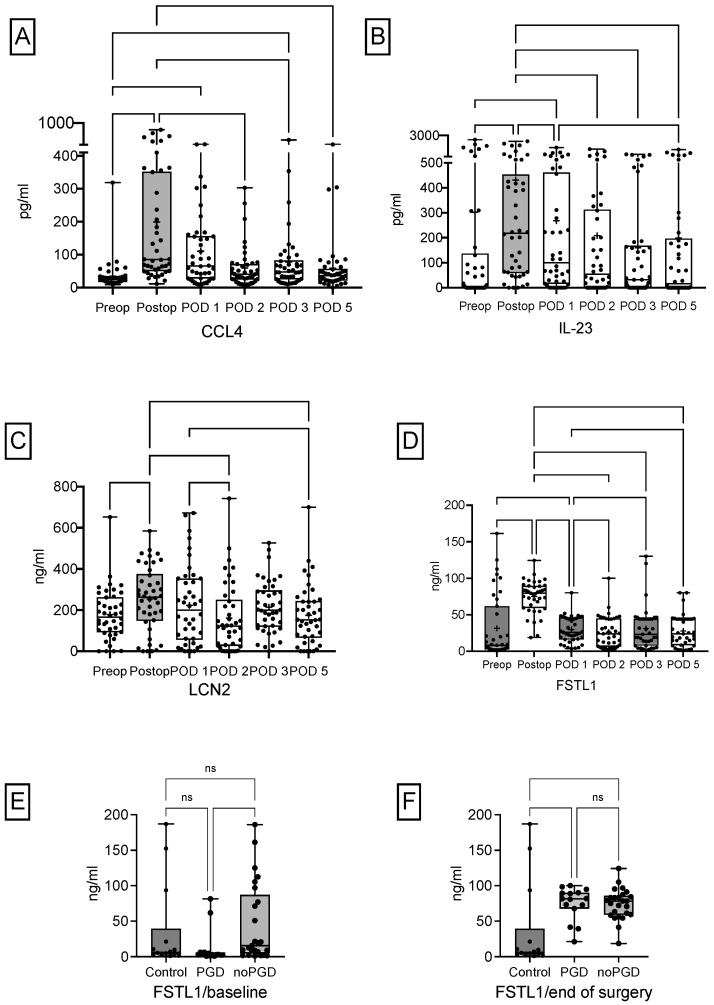
Perioperative course of cytokines. The perioperative course of (**A**) CCL4, (**B**) IL-23, (**C**) LCN2, and (**D**) FSTL1 serum concentrations of all patients undergoing LuTX is depicted. Baseline FSTL1 serum concentrations were significantly reduced in future PGD patients compared to patients without PGD (**E**). There were no differences in FSTL1 serum concentrations at the end of surgery between patients with PGD and without PGD, but in comparison to healthy volunteers without surgery (**F**). CCL4, chemokine 4; FSTL1, follistatin-like 1; IL-23, interleukin 23; LCN2, lipocalin 2; LuTX, lung transplantation; ns, not significant.

**Figure 2 biology-11-01475-f002:**
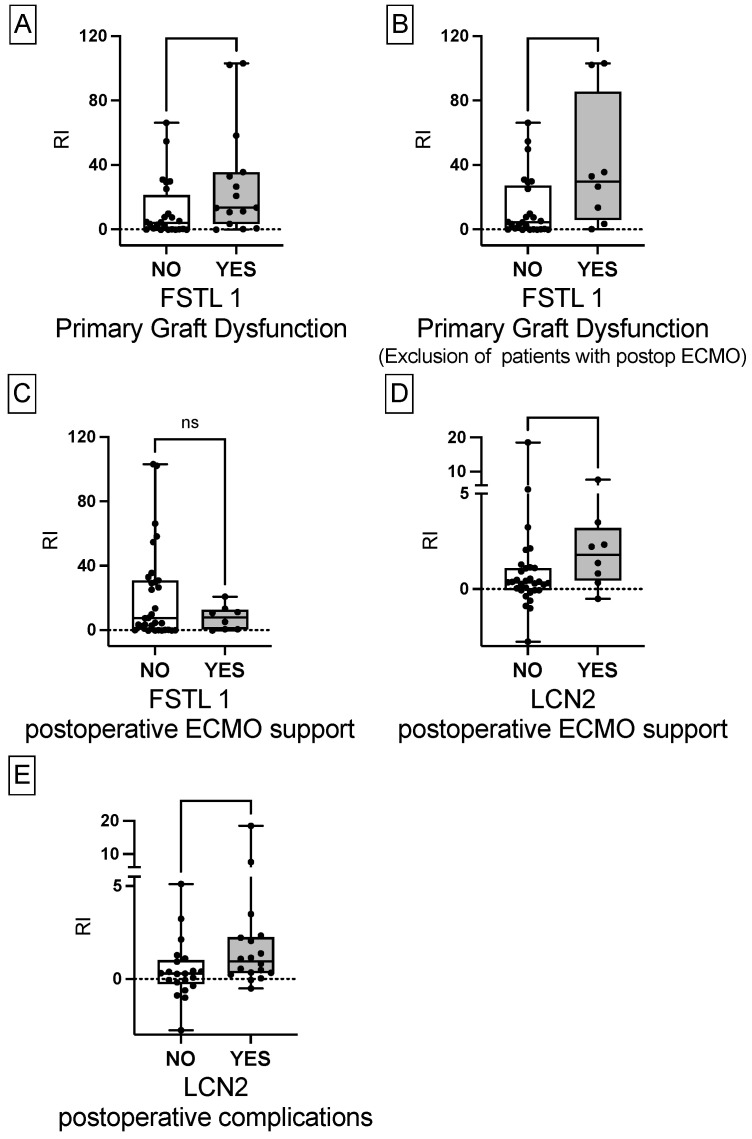
Increased cytokine serum concentrations in patients requiring prolonged ECMO, developing complications and PGD. (**A**) Significantly increased FSTL1 RI in patients developing PGD. (**B**) Significantly increased FSTL1 RI in patients developing PGD after exclusion of patients on postoperative ECMO support. (**C**) No difference in FSTL1 RI between patients with compared to patients without postoperative ECMO support. (**D**) Significantly increased LCN2 RI in patients with prolonged postoperative ECMO support. (**E**) Significantly increased LCN2 RI in patients with complications comprising hemofiltration, re-intubation, surgical revision, and 30-day mortality. ECMO, extracorporeal membrane oxygenation; FSTL1, follistatin-like 1; LCN2, lipocalin 2; ns, not significant; PGD, primary graft dysfunction; relative increase, RI.

**Figure 3 biology-11-01475-f003:**
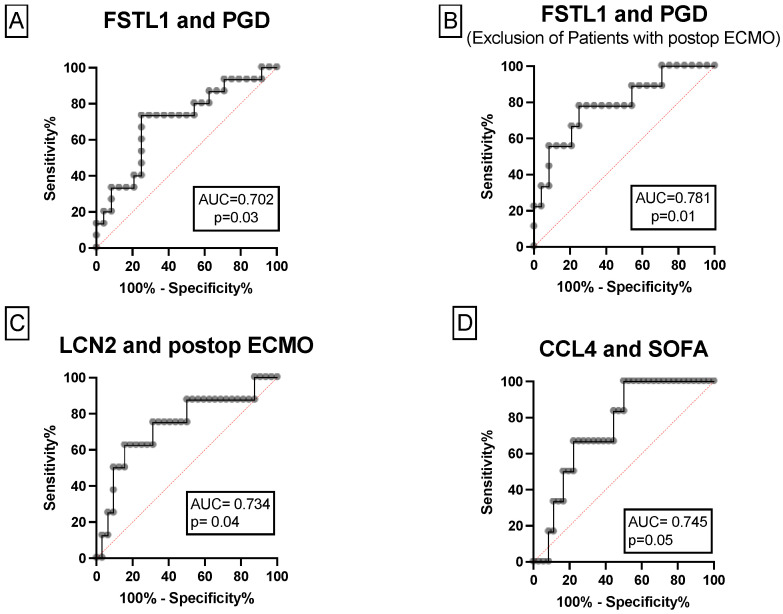
ROC curves of LCN2, CCL4 and FSTL 1. (**A**,**B**) FSTL1 FI discriminates between patients with and without PGD after LuTX even after exclusion of patients requiring postoperative ECMO support. (**C**) LCN2 FI discriminates between patients with and without prolonged ECMO support. (**D**) CCL4 FI discriminates between patients with a clinically increased inflammatory status, tangible as a SOFA score >13 and those with a SOFA < 13. CCL4, chemokine 4; ECMO, extracorporeal membrane oxygenation; FI, fold increase; FSTL1, follistatin-like 1; LCN2, lipocalin 2; LuTX, lung transplantation; PGD, primary graft dysfunction; ROC, receiver operating curve; SOFA, sequential organ failure assessment score.

**Figure 4 biology-11-01475-f004:**
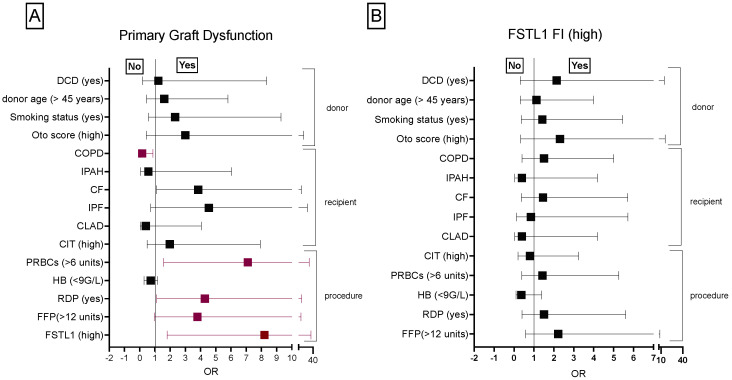
Association between procedure-, recipient- and donor-related risk factors to PGD (**A**) and high FSTL1 FI (**B**). The odds ratio was calculated for donor-, recipient- and procedure-related factors. The cut-off value for FSTL1 RI was set at 10.2. COPD, chronic obstructive pulmonary disease; DCD, donation after circulatory death; FFP, fresh frozen plasma; FSTL1, follistatin-like 1; HB, hemoglobin minimum; PRBCs, packed red blood cells; RDP, random donor platelets, RI relative increase.

**Table 1 biology-11-01475-t001:** Basic demographic data. Demographics and clinical data according to indication for LUTX are detailed below.

Diagnosis	COPD	CF	IPF	IPAH	CLAD
**Basic demographic data**					
n (%)	15 (100)	12 (100)	6 (100)	4 (100)	5 (100)
age (yrs), mean ± SD	59 ± 6	27 ± 7	54 ± 12	39 ± 12	37 ± 5
female:male ratio n	4:11	7:5	2:4	2:2	3:2
**ECMO support**					
ECMO preoperative n (%)	-	2 (17)	2 (33)	-	1 (20)
ECMO Intraoperative n (%)	15 (100)	12 (100)	6 (100)	4 (100)	5 (100)
ECMO postoperative n (%)	2 (13)	2 (17)	2 (33)	1 (25)	1 (20)
**Intraoperative characteristics**					
**Time (min)**					
Length of ECC, mean ± SD	183 ± 31	193 ± 50	174 ± 60	236 ± 39	237 ± 77
Length of surgery, mean ± SD	315 ± 70	312 ± 74	357 ± 57	380 ± 47	369 ± 131
**Vasoactive administration**					
Norepinephrine n (%)					-
<0.1 μg/kg/min	11 (73)	6 (50)	1 (17)	2 (50)	4 (80)
>0.1–0.5 μg/kg/min	4 (27)	6 (50)	5 (83)	-	1 (20)
>0.5 μg/kg/min	-	-	-	2 (50)	-
**Blood and coagulation products**					
RDP n (%)	13 (86)	6 (50)	2 (33)	1 (25)	4 (80)
PRBCs, mean ± SD	5 ± 3	9 ± 9	6 ± 3	12 ± 12	7 ± 3
FFPs, mean ± SD	11 ± 5	14 ± 12	9 ± 5	20 ± 18	10 ± 3
Max. BL (mg/dl), mean ± SD	3.1 ± 1.2	3.5 ± 1.3	3.8 ± 0.9	4.2 ± 2.1	3.6 ± 0.3
Min. HB (g/dl), mean ± SD	9.1 ± 0.9	9.2 ± 1.5	9.6 ± 1.2	8.5 ± 2.2	9.3 ± 1.3

BL, blood lactate concentration; bpm, beats per minute; CF, cystic fibrosis; CLAD, chronic lung allograft dysfunction; COPD, chronic obstructive pulmonary disease; CPB, cardiopulmonary bypass; ECC, extracorporeal circulation; ECMO, extracorporeal membrane oxygenation; FFP, fresh frozen plasma; HB, hemoglobin; HR, heart rate; IPAH, idiopathic pulmonary hypertension; IPF, idiopathic pulmonary fibrosis; LuTX, lung transplantation; n, number; POS, psycho-organic syndrome; PRBCs, packed red blood cells; RDP, random donor platelets; SD, standard deviation.

**Table 2 biology-11-01475-t002:** Grading of primary graft dysfunction. PGD grading according to the ISHLT consensus definition of 2017 at 0 h, 24 h, 48 h, and 72 h after surgery. PGD on postoperative ECMO support was graded 3 when pulmonary infiltrates on chest X-ray were present; without pulmonary infiltrates it was termed “ungradable”.

	COPD	CF	IPF	IPAH	CLAD
**PGD-Grading (0 h** **) n (%** **)**					
0	13 (86)	5 (41)	2 (33)	3 (75)	5 (100)
1	-	1 (8)	-	-	
2	-	1 (8)	2 (33)	-	-
3	-	2 (16)	-	-	-
ECMO (3)	1 (7)	1 (8)	1 (16)	1 (25)	-
ECMO (ungradable)	1 (7)	2 (16)	1 (16)		-
Extubated	-	-	-	-	-
**PGD-Grading (24 h** **) n (%** **)**					
0	7 (47)	5 (41)	3 (50)	3 (75)	2 (60)
1	-	1 (8)	-	-	1 (20)
2	-	1 (8)	-	-	
3	-	1 (8)	-		
ECMO (3)	1 (7)	1 (8)	1 (16)	1 (25)	
ECMO (ungradable)	1 (7)	-	1 (16)	-	
Extubated	6 (40)	4 (33)	1 (16)	-	1 (20)
**PGD-Grading (48 h** **) n (%** **)**					
0	3 (20)	3 (25)	1 (16)	4 (80)	1 (20)
1	1 (7)	-	1 (16)	-	
2	-	-	1 (16)	-	
3	-	-	-	-	
ECMO (3)	-	1 (8)	-	-	
ECMO (ungradable)	-	-	1 (16)	-	
Extubated	11 (73)	8 (66)	2 (33)	-	4 (80)
**PGD-Grading (72 h** **) n (%** **)**					
0	2 (13)	2 (16)	2 (33)	2 (50)	-
1	-	-	-	-	-
2	-	1 (8)	-	-	-
3	-	-	-	-	-
ECMO (3)	-	-	-	-	-
ECMO (ungradable)	-	-	1 (16)	-	-
Extubated	13 (87)	9 (75)	4 (50)	2 (50)	5 (100)

CF, cystic fibrosis; CLAD, chronic lung allograft dysfunction; COPD, chronic obstructive pulmonary disease; ECMO, extracorporeal membrane oxygenation; IPAH, idiopathic pulmonary hypertension; IPF, idiopathic pulmonary fibrosis; n, number; PGD, primary graft dysfunction.

**Table 3 biology-11-01475-t003:** Short- and long-term outcome of patients following lung transplantation.

Diagnosis	COPD	CF	IPF	IPAH	CLAD
High SOFA, n (%)	-	1 (8)	1 (16)	2 (50)	2 (20)
PGD, n (%)	2 (13)	7 (58)	4 (66)	1 (25)	-
AMR, n (%)	-	1 (8)	-	-	-
Re-intubation, n (%)	2 (13)	-	-	-	-
OBS, n (%)	2 (13)	-	2 (33)	1 (25)	1 (20)
HF, n (%)	1 (6)	1 (8)	-	1 (25)	2 (20)
CLAD, n (%)	2 (13)	2 (16)	1 (16)	1 (25)	1 (20)
BOS, n (%)	2 (13)	1 (8)	-	-	-
RAS, n (%)	-	-	-	-	-
Mixed, n (%)	-	1 (8)	1 (16)	1 (25)	1 (20)
30-day mortality, n (%)	1(6)	-	-	-	1 (20)
3-year mortality, n (%)	6 (40)	2 (16)	2 (33)	-	4 (80)

AMR, antibody-mediated rejection; BOS, bronchiolitis obliterans syndrome; CF, cystic fibrosis; CLAD, chronic lung allograft dysfunction; COPD, chronic obstructive pulmonary disease; HF, hemofiltration; IPAH, idiopathic pulmonary hypertension; IPF, idiopathic pulmonary fibrosis; LuTX, lung transplantation; OBS, organic brain syndrome; PGD, primary graft dysfunction; RAS, restrictive allograft syndrome; SOFA, sequential organ failure assessment score.

## Data Availability

All data generated or analyzed during this study are included in this published article.

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
