# Peer review of "Follistatin-like 1 and Biomarkers of Neutrophil Activation Are Associated with Poor Short-Term Outcome after Lung Transplantation on VA-ECMO"

_biology, 2022, doi:10.3390/biology11101475_

Round 1

Reviewer 1 Report

Manuscript by Veraar Et al " Follistatin-like 1 and biomarkers of neutrophil activation are associated with poor short-term outcome after lung transplantation on VA-ECMO"  determined the role of FSTL1 as a biomarker in the development of PGD. 

Minor comments need to be address- 

1. Please rewrite the sentence from 353-375. It is tool long.

2. There are so many grammatical errors through out the manuscript. 

3. Figure-1 an 2 , P value ( **) are missing on graphs though mentioned in legend. 

Author Response

Reviewer 1:

Manuscript by Veraar Et al " Follistatin-like 1 and biomarkers of neutrophil activation are associated with poor short-term outcome after lung transplantation on VA-ECMO” determined the role of FSTL1 as a biomarker in the development of PGD. 

Comment1a:

Please rewrite the sentence from 353-375. It is tool long.

Reply1a:

Dear Reviewer, thank you for this comment. We shortened the sentence as shown below.

Change1a:

In this study on patients with end-stage pulmonary disease undergoing LuTX on intraoperative ECMO support we observed significantly reduced FSTL1 serum concentrations in future PGD patients prior to LUTX compared to patients who did not develop PGD.

Comment2a:

There are so many grammatical errors through out the manuscript. 

Reply2a:

Dear reviewer, thank you for this remark we revised the manuscript and corrected following mistakes as listed below:

Changes2a:

Discussion (P.13)

Immediately after surgery, FSTL1 serum concentrations dropped back to baseline levels again in both groups.These findings suggesting that there are preoperative factors may limit an adequate FSTL1 production and therefore put these patients at an increased risk for PGD.

Research reported that in the early stage of fibrotic diseases, FSTL1 serum concentrations were up regulated.

(P.14)

Donor-specific blood transfusion given before transplantation, however, can obviously beneficially mediate allograft tolerance via a FSTL1 pathway.

(P.15)

The FSTL1 levels in asthmatic patients’ serum and bronchoalveolar lavage fluid were higher than in healthy controls, and the degree of its augmentation was positively correlated with tracheal airway smooth muscle and reticular basement membrane thickening.

Comment3a:

Figure-1 an 2 , P value ( **) are missing on graphs though mentioned in legend. 

Reply 3a:

Dear Reviewer, thank you for this comment. Symbols did not remain during the transition from word into PDF.  We re-added the Figures containing symbols into the document.

Changes3a:

Figure 1

Figure 2

Reviewer 2 Report

The authors should be congratulated on their analysis of the longitudinal perioperative course of cytokines related to neutro-18 phil activation and a cytokine that has been implicated in graft-versus-host 19 disease in 42 consecutive patients undergoing LTX. However, I do not believe that this specific topic of PGD in lung transplant recipients would be of interest to readership of Biology journal.

Therefore, I would suggest the authors to re-submit to the Special Issue "Current trends in lung transplantation" in the alternate journal Life. 

Author Response

Dear Reviewer 2, 

Thank you for appreciating our manuscript. 

We added the final version of this work below. 

Best regards ,

Cecilia Veraar and Bernhard Moser 
